# Development and validation of a 30-day mortality index based on pre-existing medical administrative data from 13,323 COVID-19 patients: The Veterans Health Administration COVID-19 (VACO) Index

**Joseph T. King, Jr.**[1,2], **James S. Yoon**[1,2,3], **Christopher T. Rentsch**[1,4], **Janet P. Tate**[1,5], **Lesley S. Park**[6], **Farah Kidwai-Khan**[1,5], **Melissa Skanderson**[1], **Ronald G. Hauser**[1,7], **Daniel A. Jacobson**[8,9,10], **Joseph Erdos**[1,11], **Kelly Cho**[12,13,14], **Rachel Ramoni**[15], **David R. Gagnon**[2,16], **Amy C. Justice**[1,5,17]*

1 VA Connecticut Healthcare System, U.S. Department of Veterans Affairs, West Haven, Connecticut, United States of America, 2 Department of Neurosurgery, Yale School of Medicine, New Haven, Connecticut, United States of America, 3 Yale School of Medicine, New Haven, Connecticut, United States of America, 4 Faculty of Epidemiology and Population Health, London School of Hygiene and Tropical Medicine, London, United Kingdom, 5 Department of Internal Medicine, Yale School of Medicine, New Haven, Connecticut, United States of America, 6 Stanford Center for Population Health Sciences, Stanford University School of Medicine, Stanford, California, United States of America, 7 Department of Laboratory Medicine, Yale School of Medicine, New Haven, Connecticut, United States of America, 8 Oak Ridge National Laboratory, Biosciences Division, Oak Ridge, Tennessee, United States of America, 9 The Bredesen Center for Interdisciplinary Research and Graduate Education, University of Tennessee Knoxville, Knoxville, Tennessee, United States of America, 10 Department of Psychology, University of Tennessee Knoxville, Knoxville, Tennessee, United States of America, 11 Department of Psychiatry, Yale School of Medicine, New Haven, Connecticut, United States of America, 12 VA Boston Healthcare System, U.S. Department of Veterans Affairs, Boston, Massachusetts, United States of America, 13 Department of Medicine, Harvard Medical School, Boston, Massachusetts, United States of America, 14 Division of Aging, Brigham and Women's Hospital, Boston, Massachusetts, United States of America, 15 Office of Research and Development, Veterans Health Administration, United States Department of Veterans Affairs, Washington, DC, United States of America, 16 Department of Biostatistics, Boston University School of Public Health, Boston, Massachusetts, United States of America, 17 Yale School of Public Health, New Haven, Connecticut, United States of America

* amy.justice@yale.edu

## Abstract

### Background

Available COVID-19 mortality indices are limited to acute inpatient data. Using nationwide medical administrative data available prior to SARS-CoV-2 infection from the US Veterans Health Administration (VA), we developed the VA COVID-19 (VACO) 30-day mortality index and validated the index in two independent, prospective samples.

### Methods and findings

We reviewed SARS-CoV-2 testing results within the VA between February 8 and August 18, 2020. The sample was split into a development cohort (test positive between March 2 and April 15, 2020), an early validation cohort (test positive between April 16 and May 18, 2020), and a late validation cohort (test positive between May 19 and July 19, 2020). Our logistic

**Data Availability Statement:** The United States Department of Veterans Affairs (VA) places legal restrictions on access to veteran's health care data, which includes both identifying data and sensitive patient information. The analytic data sets used for this study are not permitted to leave the VA firewall without a Data Use Agreement. This limitation is consistent with other studies based on VA data. However, VA data are made freely available to researchers behind the VA firewall with an approved VA study protocol. For more information, please visit https://www.virec.research.va.gov or contact the VA Information Resource Center (VIReC) at VIReC@va.gov.

**Funding:** ACJ: Department of Veterans Affairs, Office of Research and Development, Million Veteran Program Core (#MVP000; https://www. research.va.gov/). ACJ: National Institute on Alcohol Abuse and Alcoholism (U01-AA026224, U24-AA020794, U01-AA020790, U10-AA013566; https://www.niaaa.nih.gov/). DAJ: Laboratory Directed Research and Development Program of Oak Ridge National Laboratory, managed by UT-Battelle, LLC for the US Department of Energy (LOIS:10074). The views and opinions expressed in this manuscript are those of the authors and do not necessarily represent those of the Department of Veterans Affairs, Department of Energy, or the United States Government. The funders had no role in study design, data collection and analysis, decision to publish, or preparation of the manuscript.

**Competing interests:** The authors have declared that no competing interests exist.

regression model in the development cohort considered demographics (age, sex, race/ethnicity), and pre-existing medical conditions and the Charlson Comorbidity Index (CCI) derived from ICD-10 diagnosis codes. Weights were fixed to create the VACO Index that was then validated by comparing area under receiver operating characteristic curves (AUC) in the early and late validation cohorts and among important validation cohort subgroups defined by sex, race/ethnicity, and geographic region. We also evaluated calibration curves and the range of predictions generated within age categories. 13,323 individuals tested positive for SARS-CoV-2 (median age: 63 years; 91% male; 42% non-Hispanic Black). We observed 480/3,681 (13%) deaths in development, 253/2,151 (12%) deaths in the early validation cohort, and 403/7,491 (5%) deaths in the late validation cohort. Age, multimorbidity described with CCI, and a history of myocardial infarction or peripheral vascular disease were independently associated with mortality–no other individual comorbid diagnosis provided additional information. The VACO Index discriminated mortality in development (AUC = 0.79, 95% CI: 0.77–0.81), and in early (AUC = 0.81 95% CI: 0.78–0.83) and late (AUC = 0.84, 95% CI: 0.78–0.86) validation. The VACO Index allows personalized estimates of 30-day mortality after COVID-19 infection. For example, among those aged 60–64 years, overall mortality was estimated at 9% (95% CI: 6–11%). The Index further discriminated risk in this age stratum from 4% (95% CI: 3–7%) to 21% (95% CI: 12–31%), depending on sex and comorbid disease.

## Conclusion

Prior to infection, demographics and comorbid conditions can discriminate COVID-19 mortality risk overall and within age strata. The VACO Index reproducibly identified individuals at substantial risk of COVID-19 mortality who might consider continuing social distancing, despite relaxed state and local guidelines.

## Introduction

The highly contagious nature of severe acute respiratory syndrome coronavirus 2 (SARS-CoV-2), the lack of widespread immunity, and the absence of an effective vaccine ensure continued spread of the virus among the general population [1]. As state and local authorities relax guidelines, we need accurate and reliable means of identifying those at greatest risk should they become infected to inform personal choice and public policy.

Several studies have identified risk factors for mortality associated with coronavirus disease 2019 (COVID-19) in the inpatient setting [2–7]. However, these analyses do not adequately address the issue of identifying at-risk individuals before infection, for several reasons. First, these analyses were not exclusively based on data present prior to SARS-CoV-2 infection. Second, the models require data not routinely available or directly analyzable from administrative databases or electronic health records (EHR) making them difficult to apply in real time to large patient populations. Third, a recent systematic review [4] found that most SARS-CoV-2 infection outcome models were based on limited sample sizes, were likely over-fit, and were not validated in independent data.

The Veterans Health Administration (VA) is the largest integrated health care system in the United States, providing care at 1,255 health care facilities, including 170 medical centers and 1,074 outpatient sites, serving 6 million Veterans each year. Using data routinely available and

directly analyzable in the VA national system, we developed the VA COVID-19 (VACO) Index estimating 30-day COVID-19 mortality after a positive test based on demographics and pre-existing conditions, and validated its discrimination and calibration. We explored the VACO Index performance in two different time intervals of the pandemic, and in important clinical subgroups by sex, race/ethnicity, geographic region, and within age strata.

## Methods

### Data source and participants

We obtained individual patient data on August 19, 2020 from the VA Corporate Data Warehouse, which includes daily updates from over 1,200 facilities across the United States. All Veterans who were alive as of January 1, 2020 and active in care (defined as having at least one clinical encounter between January 1, 2018 and December 31, 2019, with either a recorded blood pressure or a routine laboratory test result (complete blood count, serum creatinine, alanine transaminase, or aspartate aminotransferase) were eligible. We included patients who tested positive for SARS-CoV-2 in inpatient or outpatient settings between March 2 and July 18, 2020 and followed them for 30 days.

We identified tested individuals using text searches of laboratory results containing terms consistent with SARS-CoV-2 or COVID-19. Nearly all tests utilized nasopharyngeal swabs; <1% were from other sources, serum tests were excluded. Testing was performed in VA, state public health, and commercial reference laboratories using emergency use authorization approved SARS-CoV-2 assays. If an individual had more than one test, we used the date of their first positive test. Baseline was defined as the date of specimen collection unless testing occurred during hospitalization, in which case it was defined as date of admission. If admission began more than 14 days prior to testing, possibly indicating nosocomial infection, we set the baseline to 14 days prior to testing to delineate health status before SARS-CoV-2 infection.

The data were split into a development cohort (positive test between March 2 and April 15, 2020), an early validation cohort (positive test between April 16 and May 18, 2020), and a late validation cohort (positive test between May 19 and July 19, 2020). Date of last follow-up was August 18, 2020 to allow 30 days of follow-up after testing for all patients. This study was conducted in compliance with the Health Insurance Portability and Accountability Act (HIPAA) and was approved by the Institutional Review Boards of VA Connecticut Healthcare System and Yale University, both of whom granted wavers of consent. This cohort study is reported as per the Strengthening the Reporting of Observational Studies in Epidemiology (STROBE) guidelines (S1 Checklist).

### VACO Index development: Candidate predictors

We began by performing a literature review to identify candidate demographic and medical condition predictors available in medical administrative records. Demographic variables included age, sex (male or female), and race and ethnicity (non-Hispanic Black, non-Hispanic White, Hispanic, or other). Medical conditions included individual components of the Charlson Comorbidity Index (CCI) and the CCI without an age adjustment derived from International Classification of Diseases, 10th edition (ICD-10) codes [8, 9] present between 730 and 15 days before COVID-19 testing (S1 Table). Using a previously validated grouping of ICD-10 code-defined comorbidities recorded during at least one inpatient or two outpatient encounters within the past two years [10, 11], we also considered conditions reported by other investigators as associated with COVID-19 mortality that were not included in CCI: asthma and hypertension [12–14].

Deaths were determined using inpatient records and the VA death registry to capture deaths occurring outside hospitalization. Previous research has demonstrated that these combined sources are as accurate and more up to date than the National Death Index [15].

## Statistical analyses

All data analyses were performed using Stata, version 15.1 (StataCorp, College Station, TX). We assessed the distribution of variables in the development cohort and their association and functional forms with 30-day mortality using unadjusted and multivariable logistic regression models. All variables with P<0.1 in unadjusted models were evaluated for inclusion in the adjusted models and retained in the final adjusted model for a P<0.05. We double checked the final multivariable model by reinserting and assessing the significance of previously excluded individual comorbidity and condition variables–none attained significance at P<0.05. Sex was included in the final multivariable model, regardless of P-value. CCI values with similar mortality rates were collapsed into five categories (0, 1–3, 4–5, 6–9, 10+). We explored interactions between variables—there was a significant interaction between age and CCI below the age of 85 that was incorporated into the final model.

## Model validation and calibration

We report area under the receiver operating characteristic curves (AUC) and calibration curves as assessments of the VACO Index performance in development and validation samples. To validate performance, we froze statistical weights from the final development model, then generated risk prediction scores for individuals in validation. We used the early and late validation cohorts, and a combined validation cohort, to evaluate Index performance overall and in important subgroups: sex (male vs female), race/ethnicity (Black vs non-Black), and VA-defined geographic regions combined to generate two approximately equal population samples (Northeast and West vs Southeast and Midwest). We assessed Index calibration with the Hosmer-Lemeshow goodness-of-fit test in the development cohort, and with plots of observed versus predicted 30-day mortality in 10 strata containing equal numbers of deaths, in development and validation cohorts and in validation cohort subgroups by sex, race/ethnicity, and geographic region. We also compared the range of predicted mortality values stratified by age category.

## Results

### Participants

Among tests performed from February 8 to July 19, 2020, we identified 13,323 individuals testing positive for SARS-CoV-2 in the VA who met our inclusion criteria. The first VA positive test was on March 2, 2020. Based on date of their first positive test, we assigned 3,681 patients to the development cohort, 2,151 patients to the early validation cohort, and 7,491 patients to the late validation cohort (Fig 1). As of August 18, 2020, we observed 1,136 deaths (9%): 480 (13%) in the development cohort, 253 (12%) in the early validation cohort, and 403 (5%) in the late validation cohort. The development cohort was older (median age: 64.8 vs 62.3), with a higher proportion of non-Hispanic Blacks (52% vs 38%), and a lower proportion of males (93% vs 90%) than the combined validation cohorts (Table 1). The development cohort had fewer patients with a Charlson Comorbidity Index of zero indicating absence of comorbid disease (26% vs 35%).

### VACO Index development

Univariate analyses demonstrated strong associations between multiple candidate predictors and 30-day mortality in the development cohort (Table 2). The strongest predictor was age, with mortality ranging from 0.3% among those under age 50 to 44% among those 90 or more years of age. Women experienced lower mortality than men. Before adjustment, non-Hispanic

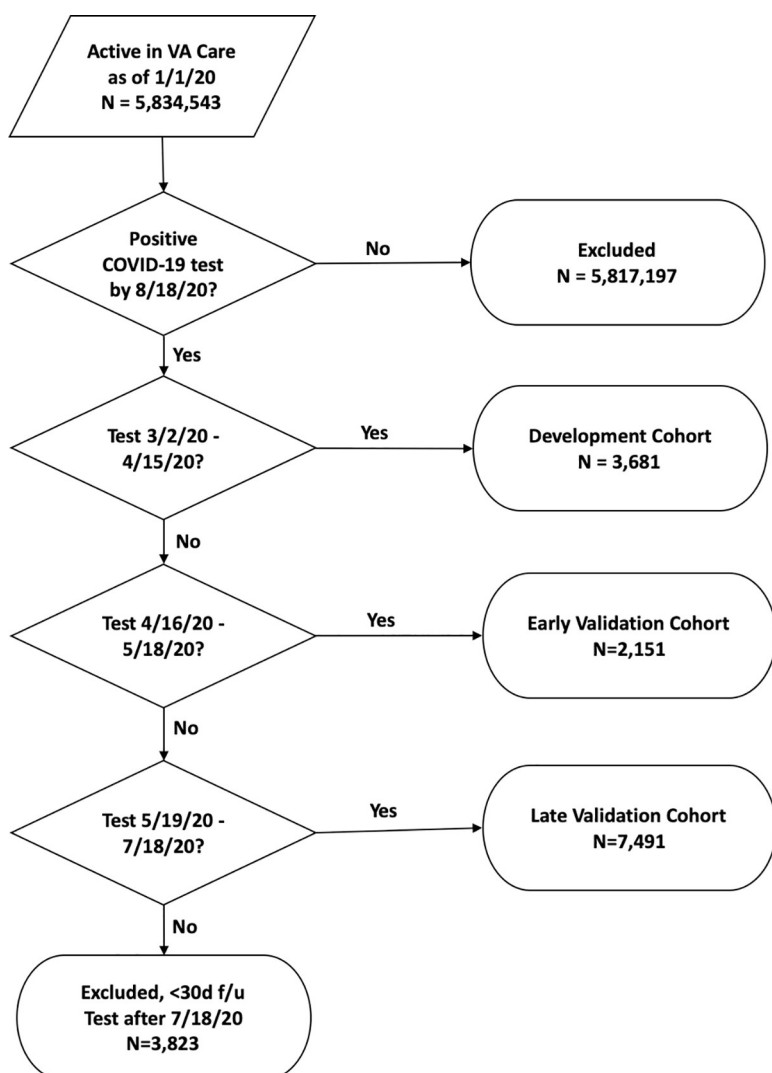

**Fig 1. Flow diagram of VACO Index cohort selection.** Flow diagram showing selection of VACO Index cohorts from 5,834,543 patients active in VA care as of January 1, 2020. All COVID-19 tests were performed in the VA. Patients with COVID-19 tests after July 18, 2020 did not have 30 days of follow-up and were excluded from the analysis.

White patients had higher mortality, although these differences vanished after adjustment with age and CCI. Many pre-existing conditions were associated with mortality including prior myocardial infarction (MI), chronic kidney disease (CKD), chronic lung disease, diabetes with complications, hypertension, and peripheral vascular disease (PVD), both individually and combined in the CCI.

## VACO Index specification and performance

Age alone was strongly associated with mortality (Table 2) with an AUC of 0.77 (95% CI: 0.75–0.79). There was a significant interaction between CCI and age below the age of 85. Discrimination improved in the multivariable model after supplementing age with sex, CCI, and MI or PVD (AUC: 0.79, 95% CI: 0.77–0.81; Fig 2). When we applied the VACO Index to the validation cohorts, it maintained good discrimination in the early (AUC: 0.81, 95% CI: 0.78–0.83) and late (AUC: 0.84, 95% CI: 0.78–0.86) validation cohorts. The AUCs for important

**Table 1. Characteristics of patients in VACO Index development and validation cohorts.**

| | Cohort | | | | | |
|---|---|---|---|---|---|---|
| | **Combined Development & Validation** | **Development** | **Validation, Early** | **Validation, Late** | **Validation, Combined** | **P value***|
| Testing dates | 3/2/2020–7/18/2020 | 3/2/2020–4/15/2020 | 4/16/2020–5/18/2020 | 5/19/2020–7/18/2020 | 4/16/2020–7/18/2020 | |
| N | 13,323 | 3,681 | 2,151 | 7,491 | 9,642 | |
| 30-day Deaths, n (%) | 1,136 (8.5) | 480 (13.0) | 253 (11.8) | 403 (5.4) | 656 (6.8) | |
| Age, median (IQR) | 63.1 (50.0–72.8) | 64.8 (53.7–73.4) | 67.6 (57.5–75.0) | 60.6 (46.0–71.7) | 62.3 (48.8–72.5) | <0.001 |
| Categories, N (%) | | | | | | |
| 20–49 | 3,326 (25.0) | 717 (19.5) | 324 (15.1) | 2,285 (30.5) | 2,609 (27.1) | <0.001 |
| 50–54 | 1,072 (8.0) | 279 (7.6) | 130 (6.0) | 663 (8.9) | 793 (8.2) | |
| 55–59 | 1,292 (9.7) | 375 (10.2) | 204 (9.5) | 713 (9.5) | 917 (9.5) | |
| 60–64 | 1,598 (12.0) | 481 (13.1) | 282 (13.1) | 835 (11.1) | 1,117 (11.6) | |
| 65–69 | 1,472 (11.0) | 433 (11.8) | 256 (11.9) | 783 (10.5) | 1,039 (10.8) | |
| 70–74 | 2,119 (15.9) | 654 (17.8) | 415 (19.3) | 1,050 (14.0) | 1,465 (15.2) | |
| 75–79 | 1,004 (7.5) | 293 (8.0) | 200 (9.3) | 511 (6.8) | 711 (7.4) | |
| 80–89 | 1,043 (7.8) | 326 (8.9) | 237 (11.0) | 480 (6.4) | 717 (7.4) | |
| ≥90 | 397 (3.0) | 123 (3.3) | 103 (4.8) | 171 (2.3) | 274 (2.8) | |
| Race/Ethnicity | | | | | | |
| Non-Hispanic White | 5,148 (38.6) | 1,194 (32.4) | 934 (43.4) | 3,020 (40.3) | 3,954 (41.0) | <0.001 |
| Non-Hispanic Black | 5,589 (42.0) | 1,896 (51.5) | 892 (41.5) | 2,801 (37.4) | 3,693 (38.3) | |
| Hispanic | 1,734 (13.0) | 405 (11.0) | 203 (9.4) | 1,126 (15.0) | 1,329 (13.8) | |
| Other/Unknown | 852 (6.4) | 186 (5.1) | 122 (5.7) | 544 (7.3) | 666 (6.9) | |
| Male sex | 12,114 (90.9) | 3,410 (92.6) | 1,993 (92.7) | 6,711 (89.6) | 8,704 (90.3) | <0.001 |
| *Comorbidity* | | | | | | |
| Asthma | 663 (5.0) | 237 (6.4) | 89 (4.1) | 337 (4.5) | 426 (4.4) | <0.001 |
| Hypertension | 7,825 (58.7) | 2,321 (63.1) | 1,424 (66.2) | 4,080 (54.5) | 5,504 (57.1) | <0.001 |
| **Charlson Comorbidities** | | | | | | |
| AIDS | 223 (1.7) | 76 (2.1) | 36 (1.7) | 111 (1.5) | 147 (1.5) | 0.033 |
| Cancer | 1,585 (11.9) | 505 (13.7) | 288 (13.4) | 792 (10.6) | 1,080 (11.2) | <0.001 |
| Cancer, metastatic | 228 (1.7) | 73 (2.0) | 45 (2.1) | 110 (1.5) | 155 (1.6) | 0.141 |
| Cerebrovascular accident | 1,578 (11.8) | 484 (13.1) | 370 (17.2) | 724 (9.7) | 1,094 (11.3) | 0.004 |
| Chronic pulmonary disease | 3,022 (22.7) | 956 (26.0) | 541 (25.2) | 1,525 (20.4) | 2,066 (21.4) | <0.001 |
| Congestive heart failure | 1,857 (13.9) | 587 (15.9) | 396 (18.4) | 874 (11.7) | 1,270 (13.2) | <0.001 |
| Diabetes | 4,900 (36.8) | 1,485 (40.3) | 874 (40.6) | 2,541 (33.9) | 3,415 (35.4) | <0.001 |
| Diabetes with complications | 2,813 (21.1) | 884 (24.0) | 544 (25.3) | 1,385 (18.5) | 1,929 (20.0) | <0.001 |
| Dementia | 1,337 (10.0) | 434 (11.8) | 368 (17.1) | 535 (7.1) | 903 (9.4) | <0.001 |
| Liver disease, mild | 1,387 (10.4) | 429 (11.7) | 274 (12.7) | 684 (9.1) | 958 (9.9) | 0.004 |
| Liver disease, severe | 140 (1.1) | 36 (1.0) | 30 (1.4) | 74 (1.0) | 104 (1.1) | 0.608 |
| Myocardial infarction | 742 (5.6) | 219 (5.9) | 172 (8.0) | 351 (4.7) | 523 (5.4) | 0.240 |
| Peptic ulcer disease | 218 (1.6) | 64 (1.7) | 47 (2.2) | 107 (1.4) | 154 (1.6) | 0.567 |
| Peripheral vascular disease | 1,800 (13.5) | 572 (15.5) | 385 (17.9) | 843 (11.3) | 1,228 (12.7) | <0.001 |
| Plegia | 276 (2.1) | 69 (1.9) | 81 (3.8) | 126 (1.7) | 207 (2.1) | 0.319 |
| Renal disease | 2,365 (17.8) | 770 (20.9) | 459 (21.3) | 1,136 (15.2) | 1,595 (16.5) | <0.001 |
| Rheumatologic disease | 243 (1.8) | 79 (2.1) | 38 (1.8) | 126 (1.7) | 164 (1.7) | 0.001 |
| Charlson Comorbidity Index | | | | | | |
| 0 | 4,321 (32.4) | 970 (26.4) | 527 (24.5) | 2,824 (37.7) | 3,351 (34.8) | <0.001 |

*(Continued)*

**Table 1.** (Continued)

| | Cohort | | | | | |
| | Combined Development & Validation | Development | Validation, Early | Validation, Late | Validation, Combined | P value* |
|---|---|---|---|---|---|---|
| 1–3 | 5,521 (41.4) | 1,597 (43.4) | 900 (41.8) | 3,024 (40.4) | 3,924 (40.7) | |
| 4–5 | 1,661 (12.5) | 517 (14.0) | 336 (15.6) | 808 (10.8) | 1,144 (11.9) | |
| 6–9 | 1,562 (11.7) | 502 (13.6) | 330 (15.3) | 730 (9.7) | 1,060 (11.0) | |
| ≥10 | 258 (1.9) | 95 (2.6) | 58 (2.7) | 105 (1.4) | 163 (1.7) | |

** Development vs Combined validation cohorts

Abbreviations: IQR = interquartile range, AIDS = acquired immunodeficiency syndrome

subgroups in the early, late, and combined validation cohorts suggested good model discrimination in men vs women, Black vs non-Black individuals, and between those living in VA Northeast and West regions vs the Southeast and Midwest regions (Table 3).

## Calibration and discrimination of the VACO Index beyond age alone

Hosmer-Lemeshow goodness-of-fit testing supported good calibration of the index in development (P = 0.847, indicating no significant lack of fit). Calibration curves of predicted versus observed 30-day mortality illustrated good calibration of the VACO Index in development, with modest overestimation of mortality in the early and late validation cohorts in which overall observed mortality rates progressively decreased (Fig 3). The VACO index demonstrated stable performance between the development and combined validation cohorts across sex, race/ethnicity, and geographic region subgroups (Fig 4).

The VACO Index can be used to estimate COVID-19 30-day mortality risk by age strata and covariates (Fig 5; S1 File). For example, among males 60–64 years of age, overall mortality was estimated as 9% (95% CI: 6–11%). The VACO Index provided risk estimates ranging from 5% (95% CI: 3–7%) for men with a CCI of zero indicating no comorbidity, to 22% (95% CI: 12–31%) for men with a CCI of 10 or more and a history of MI or PVD. Similar trends were seen across other age strata.

## Discussion

Using information present prior to SARS-CoV-2 infection from a national healthcare system, we created and validated in two prospective, independent samples a practical index that can predict 30-day COVID-19 mortality. The VACO Index is based on real world data, routinely available in medical administrative datasets. Our findings describe the experience of a large, racially and ethnically diverse, fully integrated healthcare system, encompassing inpatient and outpatient care. Discrimination of the VACO Index was maintained in both validation samples, and despite major changes in overall observed mortality over time, the Index only modestly overestimated mortality in the validation samples. The VACO Index identifies individuals at greatest risk for COVID-19 mortality, enabling patients, providers, healthcare systems, insurers, and accountable care organizations to make better informed decisions.

We are one of the first groups to use pre-existing information and multivariable modeling to generate a mortality risk index, and our findings are likely more generalizable than earlier studies [16]. Our sample was larger than most prior studies and we included patients testing positive for SARS-CoV-2 in both inpatient and outpatient settings. Most importantly, discrimination and calibration of the VACO index validated well for two different time periods in the

**Table 2. VACO Index development cohort unadjusted associations with 30-day mortality (n = 3,681; 480 deaths).**

| | Odds Ratio | 95% CI | P-value |
|---|---|---|---|
| *Demographics* | | | |
| Age, in years | | | |
| 20–49 | 0.08 | (0.02–0.39) | 0.002 |
| 50–54 | Reference | - | - |
| 55–59 | 1.60 | (0.71–3.59) | 0.254 |
| 60–64 | 2.95 | (1.41–6.14) | 0.004 |
| 65–69 | 4.83 | (2.35–9.89) | <0.001 |
| 70–74 | 7.02 | (3.51–14.03) | <0.001 |
| 75–79 | 8.22 | (4.00–16.89) | <0.001 |
| 80–89 | 14.45 | (7.15–29.21) | <0.001 |
| ≥90 | 23.48 | (11.05–49.88) | <0.001 |
| Race/Ethnicity | | | |
| Non-Hispanic White | Reference | - | - |
| Non-Hispanic Black | 0.71 | (0.58–0.88) | 0.001 |
| Hispanic | 0.58 | (0.41–0.84) | 0.003 |
| Other/Unknown | 0.52 | (0.31–0.88) | 0.015 |
| Male sex | 4.67 | (2.38–9.13) | <0.001 |
| *Comorbidity* | | | |
| Asthma | 0.85 | (0.56–1.28) | 0.437 |
| Hypertension | 2.65 | (2.09–3.35) | <0.001 |
| **Charlson Comorbidities** | | | |
| AIDS | 1.13 | (0.59–2.16) | 0.708 |
| Cancer | 1.63 | (1.27–2.09) | <0.001 |
| Cancer, metastatic | 1.46 | (0.79–2.68) | 0.224 |
| Cerebrovascular accident | 1.96 | (1.54–2.50) | <0.001 |
| Chronic pulmonary disease | 1.53 | (1.24–1.88) | <0.001 |
| Congestive heart failure | 2.32 | (1.86–2.90) | <0.001 |
| Diabetes | 1.73 | (1.43–2.10) | <0.001 |
| Diabetes with complications | 2.02 | (1.64–2.47) | <0.001 |
| Dementia | 3.25 | (2.57–4.11) | <0.001 |
| Liver disease, mild | 0.82 | (0.60–1.13) | 0.226 |
| Liver disease, severe | 3.39 | (1.69–6.83) | 0.001 |
| Myocardial infarction | 2.33 | (1.69–3.22) | <0.001 |
| Peptic ulcer disease | 1.55 | (0.82–2.93) | 0.175 |
| Peripheral vascular disease | 2.74 | (2.20–3.42) | <0.001 |
| Plegia | 1.00 | (0.49–2.03) | 0.999 |
| Renal Disease | 2.51 | (2.04–3.09) | <0.001 |
| Rheumatologic disease | 1.86 | (1.08–3.21) | 0.026 |
| Charlson Comorbidity Index | | | |
| 0 | Reference | - | - |
| 1–3 | 3.91 | (2.71–5.65) | <0.001 |
| 4–5 | 6.33 | (4.23–9.46) | <0.001 |
| 6–9 | 8.12 | (5.46–12.06) | <0.001 |
| ≥10 | 9.54 | (5.41–16.83) | <0.001 |
| Charlson Comorbidity Index and Age Interaction Term | | | |
| Age <85 | | | |

(*Continued*)

**Table 2.** (Continued)

| | Odds Ratio | 95% CI | P-value |
|---|---|---|---|
| Charlson Comorbidity Index | | | |
| 0 | Reference | - | - |
| 1–3 | 3.77 | (2.48–5.73) | <0.001 |
| 4–5 | 7.12 | (4.52–11.21) | <0.001 |
| 6–9 | 8.63 | (5.51–13.52) | <0.001 |
| ≥10 | 13.65 | (7.49–24.90) | <0.001 |
| Age 85+, any Charlson Comorbidity Index value | 25.38 | (16.17–39.82) | <0.001 |

Abbreviations: CI = confidence interval, IQR = interquartile range, AIDS = acquired immunodeficiency syndrome

pandemic, and among important subgroups including men and women, racial/ethnic minorities, and those living in different geographic regions of the US.

The strong relationship between age and COVID-19 mortality has been a consistent finding across multiple studies [17–19] and age was the strongest predictor in both unadjusted and adjusted analyses. The VACO Index allows personalized estimates of 30-day mortality after COVID-19 infection stratified by age. For example, among those aged 60–64 years, overall mortality was estimated at 9% (95% CI: 6–11%). The Index further discriminated risk in this

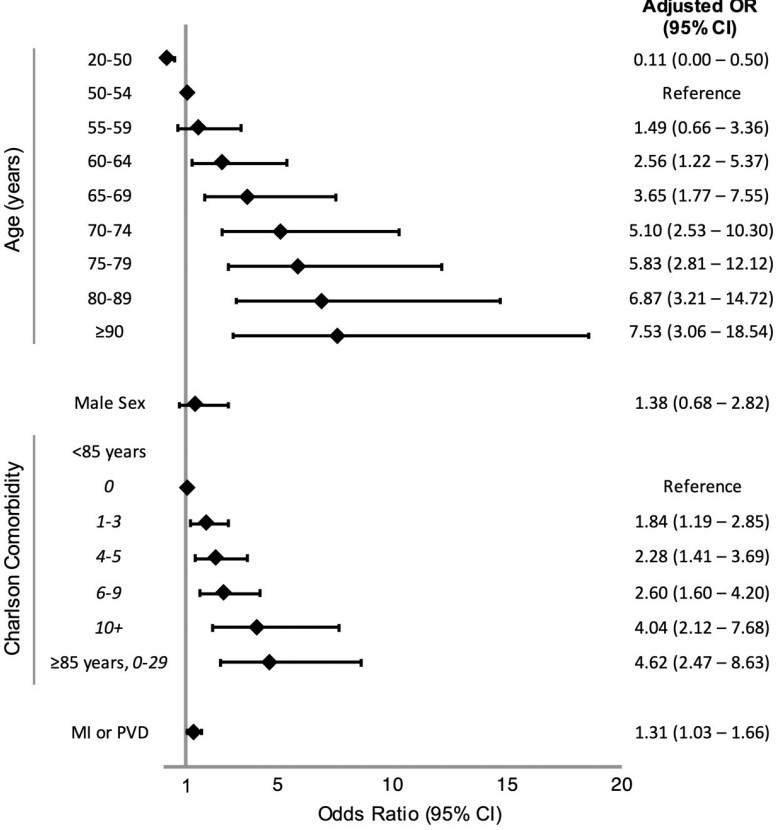

**Fig 2. Forest plot of VACO Index 30-day mortality multivariable model.** Forest plot of odds ratios (OR) and 95% confidence intervals (CI) of VACO Index variables from multivariable logistic regression model derived from development cohort (n = 3,681). Abbreviations: MI or PVD = history of myocardial infarction or peripheral vascular disease.

**Table 3. Validation of VACO Index 30-day COVID-19 mortality estimates using area under the receiver operating characteristic curves.**

| | Cohort | | | |
| --- | --- | --- | --- | --- |
| | **Development** | **Validation, Early** | **Validation, Late** | **Validation, Combined** |
| Testing Dates | 3/2/2020–4/15/2020 | 4/16/2020–5/18/2020 | 5/19/2020–7/18/2020 | 4/16/2020–7/18/2020 |
| N | 3,681 | 2,151 | 7,491 | 9,642 |
| 30-day Deaths, n (%) | 480 (13.0) | 253 (11.8) | 403 (5.4) | 656 (6.8) |
| Model, AUC (95% CI) | | | | |
| Age | 0.77 (0.75–0.79) | 0.80 (0.77–0.82) | 0.83 (0.81–0.84) | 0.82 (0.81–0.84) |
| Charlson | 0.73 (0.71–0.75) | 0.75 (0.72–0.78) | 0.78 (0.76–0.80) | 0.78 (0.76–0.80) |
| Index | 0.79 (0.77–0.81) | 0.81 (0.78–0.83) | 0.84 (0.78–0.86) | 0.84 (0.82–0.85) |
| Index Validation in Subgroups, AUC (95% CI) | | | | |
| Sex | | | | |
| Male | n/a | 0.80 (0.71–0.83) | 0.83 (0.81–0.84) | 0.83 (0.81–0.84) |
| Female | n/a | 0.79 (0.58–1.00) | 0.91 (0.82–0.99) | 0.87 (0.76–0.97) |
| Race/Ethnicity | | | | |
| Black | n/a | 0.79 (0.74–0.82) | 0.81 (0.78–0.84) | 0.81 (0.79–0.84) |
| Other | n/a | 0.81 (0.78–0.84) | 0.84 (0.83–0.86) | 0.85 (0.83–0.86) |
| Geographic region | | | | |
| Northeast & West | n/a | 0.81 (0.78–0.85) | 0.82 (0.80–0.86) | 0.84 (0.81–0.86) |
| Midwest & Southeast | n/a | 0.79 (0.74–0.83) | 0.84 (0.83–0.86) | 0.83 (0.82–0.84) |

Abbreviations: AUC = Area under receiver operating characteristic curse, CI = confidence interval

age stratum from 4% (95% CI: 3–7%) to 21% (95% CI: 12–31%), depending on sex and comorbid disease. This added discrimination is particularly relevant for patients age 60–74 who are both at substantial risk and often remain employed. Thirty-nine percent of those age 60–74 in the US are employed [20], thus accurate personalized risk estimation can better inform personal and system level decisions regarding returning to work or other group settings.

Most prior studies considered only individual comorbid conditions such as asthma, chronic lung disease, diabetes, hypertension, and vascular disease [6, 7, 12, 21–23]. Liang *et al.* found that comorbidity count predicted critical illness in hospitalized patients in China [3]. We found that multimorbidity captured by the CCI has a stronger relationship with mortality than nearly all individual comorbid conditions. After adjustment using the CCI, only a prior MI or PVD was independently associated with mortality. CCI also has the advantage of straightforward calculation from ICD-10 diagnosis codes obtained from medical administrative data, and is widely used across numerous diseases, health care systems, and populations [9]. Our finding that MI and PVD added independent prognostic information underscores the likely importance of thrombotic complications in COVID-19 [24, 25]. It stands to reason that those with pre-existing vascular disease are more susceptible to thrombosis if infected.

The most important limitation of the VACO Index is that it was developed on patients who presented for COVID-19 testing early in the pandemic, presumably because they had symptomatic disease. COVID-19 testing capacity in the US was limited early in the pandemic, and testing was reserved for patients with significant symptoms that might represent a more severe infection. While the discrimination of the VACO Index was maintained in both prospective independent validations, index predictions modestly over-estimated mortality risk in validation, particularly in the late validation cohort. Mortality rates among those testing positive for COVID-19 are decreasing as US testing capacity improves, permitting testing of more mildly symptomatic and asymptomatic people who are less likely to succumb to the disease. Overall mortality rate in our development cohort was nearly three times that found in our most recent

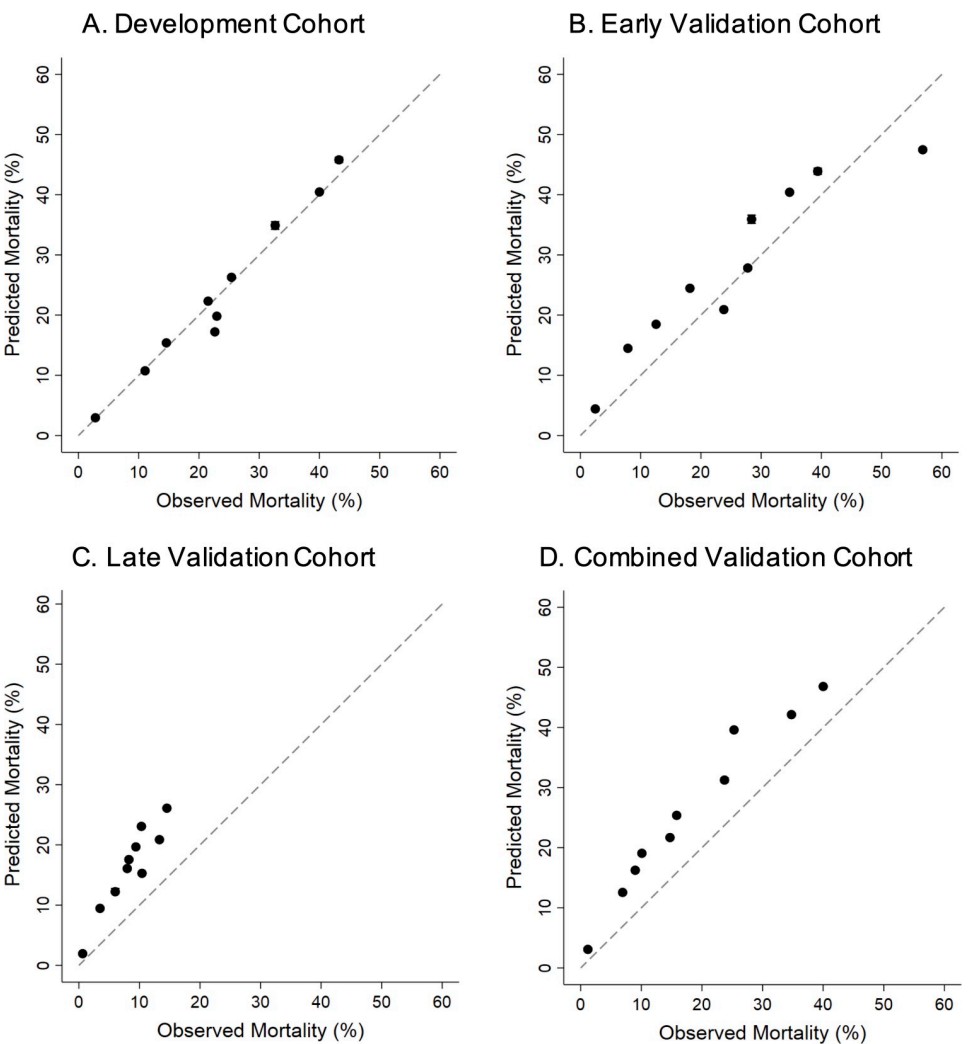

**Fig 3. Calibration plots of VACO Index: Development, early validation, late validation, and combined validation cohorts.** Calibration plots of VACO Index predicted 30-day mortality risk versus observed patient mortality across the cohorts. Error bars show 95% confidence intervals and dashed lines indicate perfect agreement between predicted versus observed patient mortality. a. Development cohort: test positive between March 2 and April 15, 2020, n = 3,681, 480 deaths. b. Early validation cohort: test positive between April 16 and May 18, 2020, n = 2,151, 253 deaths. c. Late validation cohort: test positive between May 19 and July 18, 2020, n = 7,491, 403 deaths. d. Combined early and late validation cohorts: test positive between April 16 and July 18, 2020, n = 9,642, 656 deaths.

validation cohort (13% vs 5%). Predictive indices developed in the context of high mortality rates will almost inevitably overestimate risk in samples with substantially lower mortality. However, if discrimination of the index is preserved, it is possible to adjust calibration as rates eventually stabilize.

COVID-19 testing criteria and rates, test positivity rates, and mortality are evolving with the pandemic. Centers for Disease Control and Prevention (CDC) data estimate that the number of people with antibody evidence of SARS-CoV-2 infection is many times the number of reported COVID-19 test-positive cases [26]. The CDC report did not stratify their results by age, and older people are almost certainly more likely to experience symptoms if infected. While the CDC report suggested that the overall ratio of asymptomatic to symptomatic infections was ~10:1, it may be substantially lower for older individuals. Future research should

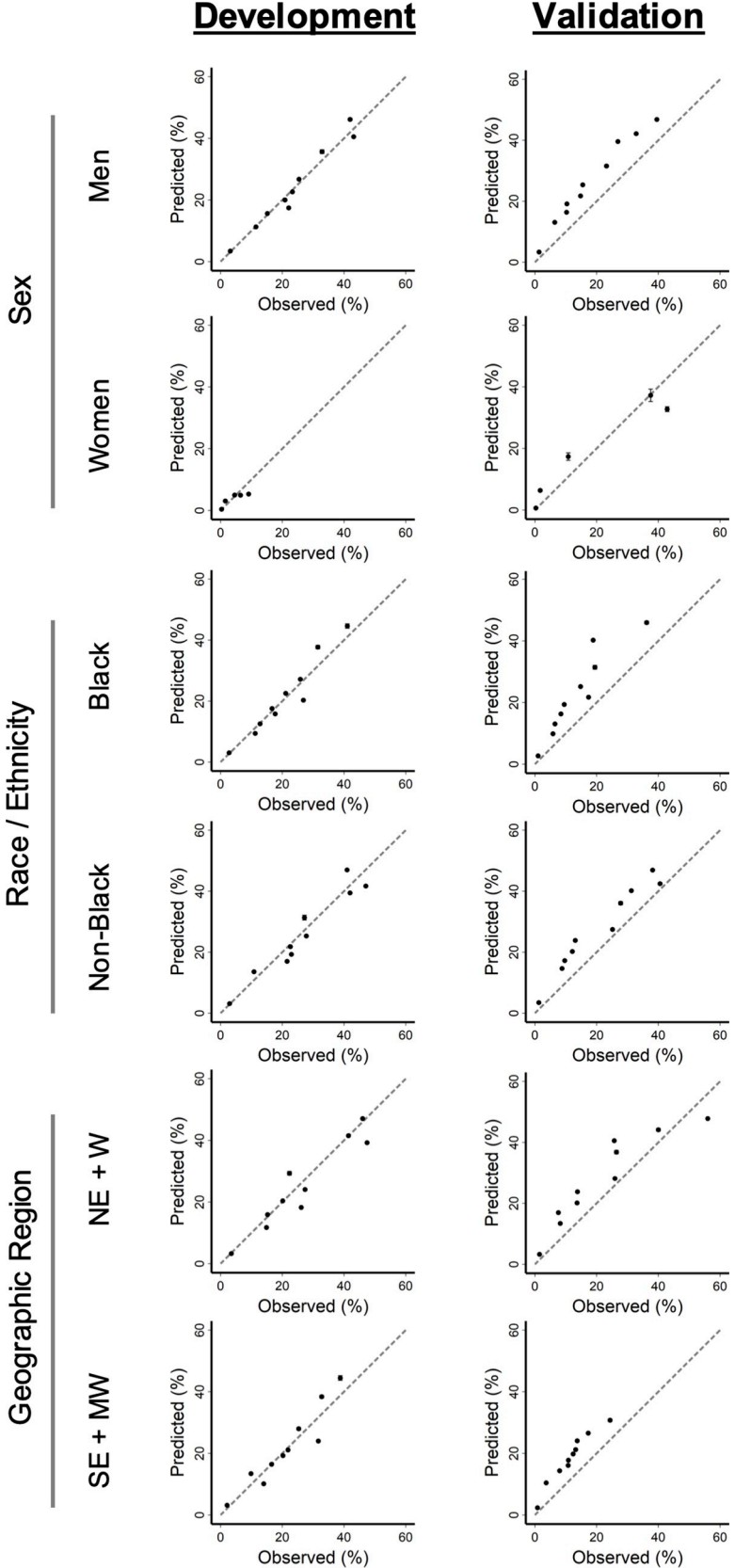

**Fig 4. Calibration plots of VACO Index: Combined cohort subgroups.** Calibration plots of VACO Index 30-day predicted mortality risk versus observed patient mortality. Error bars show 95% confidence intervals and dashed lines indicate perfect agreement between predicted versus observed patient mortality. Development cohort: test positive between March 2 and April 15, 2020, n = 3,681, 480 deaths. Combined early and late validation cohorts: test positive between April 16 and July 18, 2020, n = 9,642, 656 deaths. Subgroups: Men vs women; Black vs non-Black race; Northeast (NE) + West (W) regions vs Southeast (SE) + Midwest (MW) regions.

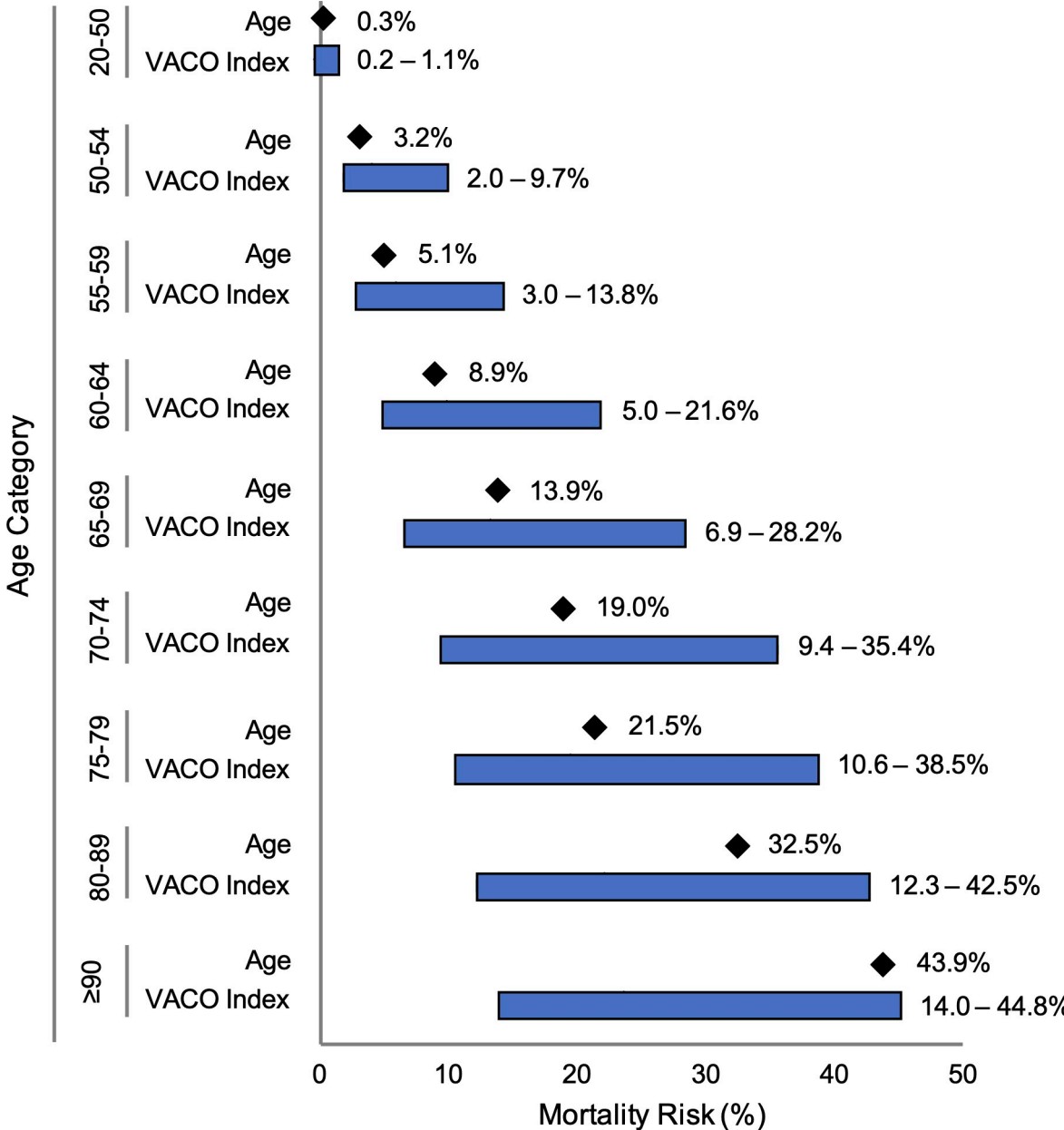

**Fig 5. Range of 30-day mortality predictions from age alone and VACO Index.** Bar graphs demonstrating the additional variation in mortality prediction provided by the VACO Index over age alone across age categories in the combined validation cohort (n = 9,642). The diamonds indicate predicted 30-day mortality within each age category when only age is used to generate the predicted value. The bars show the range of predicted 30-day mortality within the same age category provided by the VACO Index, where age is supplemented with sex and comorbidities.

examine this ratio stratified by age as a potential factor in mortality risk estimation. We are gathering data to adjust risk estimates based on the ratio of asymptomatic to symptomatic infections stratified by age; however, this is beyond the scope of this analysis.

This study has other limitations. Our study population was limited to Veterans in VA care. Prior work has demonstrated that while Veterans in VA care are older and have a higher prevalence of chronic health conditions and risk behaviors than the general US population [27–29], after adjusting for age, sex, race/ethnicity, region, and residence location, there are no significant differences in total disease burden [29]. VA has excellent mortality assessment [15], but delays in registering outpatient deaths could result in some under reporting. We only included Veterans receiving COVID-19 testing in the VA—others may have been tested and treated outside the VA. In the future, when Center for Medicare and Medicaid Services (CMS) data are available, this limitation could be addressed in Veterans age 65 and older. Our goal was to create a predictive model using pre-existing data that is available and readily analyzable in real time in most medical administrative data. Consequently, we did not consider laboratory data, vital signs, medications, or information typically residing in text notes, such as symptoms, physical exam findings, or imaging. We have demonstrated internal generalizability of the VACO Index within the VA—we recommend further validation in external datasets before applying the VACO Index outside of the VA.

In summary, using data from a national healthcare system, we developed and validated the VACO Index, a short-term mortality risk index based upon directly analyzable data available prior to infection with SARS-CoV-2. By doing so, we provide timely, quantifiable, and individualized risk estimates that successfully differentiate risk of 30-day mortality among those of similar age to better inform personal decision making and public policy as countries begin to relax lockdown guidelines.

## Supporting information

**S1 Checklist. STROBE cohort study checklist.**
(DOCX)

**S1 Table. Charlson Comorbidity Index determination from ICD-10 diagnosis codes.**
(DOCX)

**S1 File. VACO Index calculation of predicted mortality.**
(DOCX)

## Acknowledgments

The authors wish to acknowledge the work of the larger VA-DOE COVID-19 Collaboration, the Veterans who choose to get their care within the VA, and Dr. Kendall Bryant, our National Institute on Alcohol Abuse and Alcoholism Scientific Collaborator.

## Author Contributions

**Conceptualization:** Joseph T. King, Jr., Amy C. Justice.

**Data curation:** Joseph T. King, Jr., Christopher T. Rentsch, Janet P. Tate, Farah Kidwai-Khan, Melissa Skanderson, Ronald G. Hauser.

**Formal analysis:** Joseph T. King, Jr., James S. Yoon.

**Funding acquisition:** Amy C. Justice.

**Methodology:** Joseph T. King, Jr., Christopher T. Rentsch, Janet P. Tate, Amy C. Justice.

**Project administration:** Amy C. Justice.

**Visualization:** Joseph T. King, Jr., James S. Yoon.

**Writing – original draft:** Joseph T. King, Jr., Amy C. Justice.

**Writing – review & editing:** Joseph T. King, Jr., James S. Yoon, Christopher T. Rentsch, Janet P. Tate, Lesley S. Park, Farah Kidwai-Khan, Melissa Skanderson, Ronald G. Hauser, Daniel A. Jacobson, Joseph Erdos, Kelly Cho, Rachel Ramoni, David R. Gagnon, Amy C. Justice.

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
