## [Decision Letter · Decision Letter 0]

8 Oct 2020

PONE-D-20-27401

Development and validation of a 30-day mortality index based on pre-existing medical administrative data from 13,323 COVID-19 patients: the Veterans Health Administration COVID-19 (VACO) Index

PLOS ONE

Dear Dr. Justice,

Thank you for submitting your manuscript to PLOS ONE. After careful consideration, we feel that it has merit but does not fully meet PLOS ONE’s publication criteria as it currently stands. Therefore, we invite you to submit a revised version of the manuscript that addresses the points raised during the review process.

Please review comments made by the reviewers. In your revised version, kindly submit a point by point response to the reviewer queries.

We look forward to receiving your revised manuscript.

Kind regards,

Muhammad Adrish

Academic Editor

PLOS ONE

Journal Requirements:

2. In your ethics statement in the Methods section and in the online submission form, please provide additional information about the data used in your retrospective study.

Specifically, please ensure that you have discussed whether all data were fully anonymized before you accessed them and/or whether an IRB or ethics committee waived the requirement for informed consent.

If patients provided informed written consent to have data from their medical records used in research, please include this information.

3. Please include the date(s) on which you accessed the databases or records to obtain the data used in your study.

4. Please provide a table of relevant baseline characteristics of the subjects.

6. One of the noted authors is a group or consortium; VA-DOE COVID-19 Collaboration.

In addition to naming the author group, please list the individual authors and affiliations within this group in the acknowledgments section of your manuscript.

Please also indicate clearly a lead author for this group along with a contact email address.

7. Please include a separate caption for each figure in your manuscript.

8. Please include a copy of Tables 1, 2 and 3 which you refer to in your text on pages 9, 10 and 11.

Reviewers' comments:

Reviewer's Responses to Questions

**Comments to the Author**

1. Is the manuscript technically sound, and do the data support the conclusions?

Reviewer #1: Yes

Reviewer #2: Yes

2. Has the statistical analysis been performed appropriately and rigorously? 

Reviewer #1: Yes

Reviewer #2: Yes

3. Have the authors made all data underlying the findings in their manuscript fully available?

Reviewer #1: Yes

Reviewer #2: No

4. Is the manuscript presented in an intelligible fashion and written in standard English?

Reviewer #1: Yes

Reviewer #2: Yes

5. Review Comments to the Author

Reviewer #1: The study is interesting, well conducted and presented. I have only the following minor suggestions.

Results

Pag 9, lines 179-182. Please specify the two comparison groups (development cohort versus both validation cohorts combined?)

Pag 9, Line 184 use the term “univariate” instead of “bivariate”.

Pag 10, line 209. Please describe Figure 4 separately from Figure 3 and specify which subgroups were considered.

Discussion

Pag 11, line 234. Please delete Table 3.

Pag. 13, line 262. Please mention that over estimation was particularly evident in the late validation cohort.

Pag. 14. In the conclusion, please mention the necessity to validate the algorithm using other external cohorts.

Tables were not included in the Manuscript.

Reviewer #2: A rigorously executed and clearly argued study that makes excellent use of VA data resources and quantitative methods. Well done splitting data into test and validation subsets and explicitly referencing STROBE elements.

The VA has several frailty/comorbidity indexes available, and it might be interesting to compare VACO to those (e.g. JEN Frailty Index, VA Frailty Index) as well but not an obstacle to acceptance of manuscript.

The full data was not made available, but this is typical for patient data, and the authors mention VA's process for obtaining authorization to access the data. The authors did make an effort to provide the data they are able to provide, in the supplementary files.

6. PLOS authors have the option to publish the peer review history of their article (what does this mean?). If published, this will include your full peer review and any attached files.

Reviewer #1: **Yes: **Cristina Bosetti

Reviewer #2: **Yes: **Alex Bokov, Ph.D.

---

## [Author Response · Author response to Decision Letter 0]

9 Oct 2020

Also contained in cover letter.

We have updated the manuscript to ensure that it meets PLOS ONE’s style requirements.

2. In your ethics statement in the Methods section and in the online submission form, please provide additional information about the data used in your retrospective study. Specifically, please ensure that you have discussed whether all data were fully anonymized before you accessed them and/or whether an IRB or ethics committee waived the requirement for informed consent. If patients provided informed written consent to have data from their medical records used in research, please include this information.

We have updated our Methods section and online submission form to reflect that the IRB approved a waiver of consent. (line 126)

3. Please include the date(s) on which you accessed the databases or records to obtain the data used in your study.

We now state in our Methods that we accessed the VA databases on August 19, 2020. (line 103)

4. Please provide a table of relevant baseline characteristics of the subjects.

Table 1 details the baseline characteristics of subjects in the development cohort and the validation cohorts.

The United States Department of Veterans Affairs (VA) places legal restrictions on access to veteran’s health care data, which includes both identifying data and sensitive patient information. The analytic data sets used for this study are not permitted to leave the VA firewall without a Data Use Agreement. This limitation is consistent with other studies based on VA data. However, VA data are made freely available to researchers behind the VA firewall with an approved VA study protocol. For more information, please visit https://www.virec.research.va.gov or contact the VA Information Resource Center (VIReC) at VIReC@va.gov.

6. One of the noted authors is a group or consortium; VA-DOE COVID-19 Collaboration. In addition to naming the author group, please list the individual authors and affiliations within this group in the acknowledgments section of your manuscript. Please also indicate clearly a lead author for this group along with a contact email address.

We have dropped the VA-DOE COVID-19 Collaboration from the author list. The Collaboration is satisfied with the recognition of specific members in the author list.

7. Please include a separate caption for each figure in your manuscript.

 We now include a separate caption for each figure in the manuscript.

8. Please include a copy of Tables 1, 2 and 3 which you refer to in your text on pages 9, 10 and 11.

We now include Tables 1, 2, and 3 in the manuscript.

Reviewers' comments:

Reviewer's Responses to Questions

Comments to the Author

1. Is the manuscript technically sound, and do the data support the conclusions?

Reviewer #1: Yes

Reviewer #2: Yes

2. Has the statistical analysis been performed appropriately and rigorously? 

Reviewer #1: Yes

Reviewer #2: Yes

3. Have the authors made all data underlying the findings in their manuscript fully available?

Reviewer #1: Yes

Reviewer #2: No

4. Is the manuscript presented in an intelligible fashion and written in standard English?

Reviewer #1: Yes

Reviewer #2: Yes

5. Review Comments to the Author

Reviewer #1: The study is interesting, well conducted and presented. I have only the following minor suggestions.

Results

Pag 9, lines 179-182. Please specify the two comparison groups (development cohort versus both validation cohorts combined?)

The current text specifies that we are comparing the development and combine validation cohorts: “The development cohort was older (median age: 64.8 vs 62.3), with a higher proportion of non-Hispanic Blacks (52% vs 38%), and a lower proportion of males (93% vs 90%) than the combined validation cohorts (Table 1).” (underline emphasis added; lines 179-181) 

Pag 9, Line 184 use the term “univariate” instead of “bivariate”.

We have changed “Bivariate” to “Univariate.” (line 197)

Pag 10, line 209. Please describe Figure 4 separately from Figure 3 and specify which subgroups were considered.

We now discuss Figure 3 and 4 in separate sentences. The sentence for Figure 3 specifies the we are comparing the development, early, and late validation cohorts. (lines 239-242) The revised sentence for Figure 4 specifies that we are comparing the validation and combined validation cohorts: “The VACO index demonstrated stable performance between the development and combined validation cohorts across sex, race/ethnicity, and geographic region subgroups (Figure 4).” (lines 242-244)

Discussion

Pag 11, line 234. Please delete Table 3.

We have deleted the reference to Table 3. (line 295)

Pag. 13, line 262. Please mention that over estimation was particularly evident in the late validation cohort.

We now specify that over estimation was particularly evident in the late validation cohort. (line 323)

Pag. 14. In the conclusion, please mention the necessity to validate the algorithm using other external cohorts.

In the Conclusion, we now state: “We have demonstrated internal generalizability of the VACO Index within the VA - we recommend further validation in external datasets before applying the VACO Index outside of the VA.” (lines 353-355)

Tables were not included in the Manuscript.

We now include Tables 1, 2 and 3 with the manuscript.

Reviewer #2: A rigorously executed and clearly argued study that makes excellent use of VA data resources and quantitative methods. Well done splitting data into test and validation subsets and explicitly referencing STROBE elements.

The VA has several frailty/comorbidity indexes available, and it might be interesting to compare VACO to those (e.g. JEN Frailty Index, VA Frailty Index) as well but not an obstacle to acceptance of manuscript.

In future work we hope to compare the VACO Index to other indices, but these comparisons are beyond the scope of this paper.

The full data was not made available, but this is typical for patient data, and the authors mention VA's process for obtaining authorization to access the data. The authors did make an effort to provide the data they are able to provide, in the supplementary files.

6. PLOS authors have the option to publish the peer review history of their article (what does this mean?). If published, this will include your full peer review and any attached files.

We agree to publishing the peer review history of our article.

Do you want your identity to be public for this peer review? For information about this choice, including consent withdrawal, please see our Privacy Policy.

Reviewer #1: Yes: Cristina Bosetti

Reviewer #2: Yes: Alex Bokov, Ph.D.

---

## [Decision Letter · Decision Letter 1]

22 Oct 2020

Development and validation of a 30-day mortality index based on pre-existing medical administrative data from 13,323 COVID-19 patients: The Veterans Health Administration COVID-19 (VACO) Index

PONE-D-20-27401R1

Dear Dr. Justice,

We’re pleased to inform you that your manuscript has been judged scientifically suitable for publication and will be formally accepted for publication once it meets all outstanding technical requirements.

Kind regards,

Muhammad Adrish

Academic Editor

PLOS ONE

Additional Editor Comments (optional):

Reviewers' comments:

Reviewer's Responses to Questions

**Comments to the Author**

1. If the authors have adequately addressed your comments raised in a previous round of review and you feel that this manuscript is now acceptable for publication, you may indicate that here to bypass the “Comments to the Author” section, enter your conflict of interest statement in the “Confidential to Editor” section, and submit your "Accept" recommendation.

Reviewer #1: All comments have been addressed

2. Is the manuscript technically sound, and do the data support the conclusions?

Reviewer #1: Yes

3. Has the statistical analysis been performed appropriately and rigorously? 

Reviewer #1: Yes

4. Have the authors made all data underlying the findings in their manuscript fully available?

Reviewer #1: Yes

5. Is the manuscript presented in an intelligible fashion and written in standard English?

Reviewer #1: Yes

6. Review Comments to the Author

Reviewer #1: The author have addressed all my previous requests. The manuscript can threfore be accepted in the present form.

7. PLOS authors have the option to publish the peer review history of their article (what does this mean?). If published, this will include your full peer review and any attached files.

Reviewer #1: No

---

## [Editor Report · Acceptance letter]

3 Nov 2020

PONE-D-20-27401R1 

Development and validation of a 30-day mortality index based on pre-existing medical administrative data from 13,323 COVID-19 patients: The Veterans Health Administration COVID-19 (VACO) Index 

Dear Dr. Justice:

I'm pleased to inform you that your manuscript has been deemed suitable for publication in PLOS ONE. Congratulations! Your manuscript is now with our production department. 

Kind regards, 

on behalf of

Dr. Muhammad Adrish 

Academic Editor

PLOS ONE